# The Impact of Demographic, Socio-Economic and Geographic Factors on Mortality Risk among People Living with Dementia in England (2002–2016)

**DOI:** 10.3390/ijerph182413405

**Published:** 2021-12-20

**Authors:** James Watson, Frances Darlington-Pollock, Mark Green, Clarissa Giebel, Asangaedem Akpan

**Affiliations:** 1School of Environmental Sciences, The University of Liverpool, Liverpool L69 7ZT, UK; f.darlington@hotmail.co.uk (F.D.-P.); mgreen@liverpool.ac.uk (M.G.); 2Department of Primary Care and Mental Health, University of Liverpool, Liverpool L69 3GF, UK; Clarissa.Giebel@liverpool.ac.uk; 3NIHR ARC NWC, Liverpool L69 3GL, UK; 4Liverpool University Hospital NHS FT, Liverpool L7 8XP, UK; asangaedem.akpan@liverpool.ac.uk

**Keywords:** dementia, early-onset, later-onset, mortality, healthcare, inequalities, social, spatial

## Abstract

Increasing numbers of people living with dementia (PLWD), and a pressured health and social care system, will exacerbate inequalities in mortality for PLWD. There is a dearth of research examining multiple factors in mortality risk among PLWD, including application of large administrative datasets to investigate these issues. This study explored variation mortality risk variation among people diagnosed with dementia between 2002–2016, based on: age, sex, ethnicity, deprivation, geography and general practice (GP) contacts. Data were derived from electronic health records from a cohort of Clinical Practice Research Datalink GP patients in England (*n* = 142,340). Cox proportional hazards regression modelled mortality risk separately for people with early- and later- onset dementia. Few social inequalities were observed in early-onset dementia; men had greater risk of mortality. For early- and later-onset, higher rates of GP observations—and for later-onset only dementia medications—are associated with increased mortality risk. Social inequalities were evident in later-onset dementia. Accounting for other explanatory factors, Black and Mixed/Other ethnicity groups had lower mortality risk, more deprived areas had greater mortality risk, and higher mortality was observed in North East, South Central and South West GP regions. This study provides novel evidence of the extent of mortality risk inequalities among PLWD. Variance in mortality risk was observed by social, demographic and geographic factors, and frequency of GP contact. Findings illustrate need for greater person-centred care discussions, prioritising tackling inequalities among PLWD. Future research should explore more outcomes for PLWD, and more explanatory factors of health outcomes.

## 1. Introduction

There are inequalities in mortality for people living with dementia (PLWD) by various socio-economic and geographic factors [1,2]. People from the most disadvantaged socio-economic groups are most likely to have unmet care needs and experience poorer health outcomes [3]. Government policy has prioritised narrowing inequalities in access to dementia diagnosis, post-diagnostic support, treatment, and health and social outcomes [4].

PLWD are more likely than the general population to have comorbidities, and as their condition progresses, a greater need for support with activities of daily living. Increased care need, care home closures and fewer care home places, and social care funding changes, means PLWD with comorbidities are more reliant on informal carers [5]. PLWD pay more out-of-pocket for social care [6] and use healthcare services more than those without dementia [7]. Increased use of acute and unplanned healthcare is associated with greater financial cost to services and poorer outcomes for PLWD [8,9]. However, current UK health and social care funding is strained, with a reliance on the individual to fund, and the third sector to provide a substantial proportion of dementia services [10]. Appropriate health and social care can slow dementia progression, improve outcomes, benefit informal carers and maintain independence for PLWD [11,12]. UK strategies and policy recommendations are not reflected in service provision, and recent government commitments to increase dementia funding have not been enacted [13].

The number of PLWD in the UK is estimated to increase from ~920,000 in 2020, to over 1 million by 2024. The reflected financial health and social care cost is anticipated to increase threefold [14]. The majority of PLWD are aged 65 and over, and the greatest increase will be among those with severe dementia, who often have the greatest needs (for support with routine daily activities), poorest prognosis [14] and greatest service costs [15]. As dementia progresses, and health deteriorates, PLWD are likely to need greater levels of healthcare involvement, both in relation to dementia, and comorbidities [16]. A further shift towards an older population and increased numbers of PLWD, more severe symptomology and poorer health will result in greater mortality risk [1]. These factors will impact some social groups more acutely, particularly older PLWD [17]. Without additional funding and support, dementia services struggling to cope with current demand—and further impacted by the COVID-19 pandemic [18]—are unlikely to be able to effectively care for and treat increased numbers of PLWD [19]. Increased demand for healthcare will exacerbate issues with care and treatment, which will likely have a disproportionate impact [20], resulting in worse outcomes, including poorer survival and greater mortality [21]. Due to inequalities in service availability, access and quality of care, the most disadvantaged will likely be impacted disproportionately [22].

In order to address current inequalities, and prevent inequalities widening due to increased demand and struggling services, we need to employ administrative data to better support improved decision-making. Mortality is a definitive outcome, which can illustrate both ill-health and effectiveness of care and symptom management throughout the trajectory of an illness [16,23]. During times of restricted funding and service availability, inequalities in mortality and differences in life expectancy across social groups widen [24]. As such, it is critical we explore current inequalities as a matter of course to understanding how we can negate the differences experiences by people as a result of belonging to specific social groups. Electronic health records (EHR) can be employed to identify healthcare use and outcomes among specified, large patient groups, which may not be possible through other means [25,26]. Though EHRs have evidenced mortality risk among PLWD, there is limited use in evidencing social, demographic and geographic inequalities [17]. With policy-makers and service commissioners increasingly aware of the need for patients’ social context in their experience of a disease, there is an urgent need to better employ EHRs to evidence variation in outcomes for PLWD. This study addresses these evidence gaps.

The aim of this study is to examine the extent to which demographic, socioeconomic, geographic and healthcare factors are associated with mortality for PLWD, using large-scale, longitudinal EHRs.

## 2. Materials and Methods

### 2.1. Data Access and Ethical Approval

Clinical Practice Research Datalink (CPRD) collects anonymised EHR from ~2000 general practices (GPs) across the UK, with ~16 million registered patients included, representing 25% of the UK’s population. GPs apply to CPRD to register for their data to be collected, resulting in potential uneven geographic coverage. CPRD has been employed previously to investigate socio-economic and demographic factors in variation in care of physical and mental health conditions [27,28]. Data access was granted by CPRD and use of CPRD Aurum for research purposes approved by the University of Liverpool Research Ethics board (Reference: 7922). The CPRD Aurum database contains routinely-collected, longitudinal EHRs from CPRD-registered GPs, providing records of patient-GP contacts and some socio-economic and geographic variables. For reference, in this paper, a “patient-GP contact” is a single record of a discussion between or regarding a patient—whether that be face-to-face or otherwise—or record of a medication prescribed to a patient—whether that be an initial medication or repeat prescription.

### 2.2. Outcome Variable

Mortality, is a binary outcome variable, based on the presence of date of death in CPRD.

### 2.3. Explanatory Variables

CPRD Aurum contains three primary datasets for patients’ contacts with their GP: consultations, observations and drug issue (medication) records [29]. Observation records include clinical measurements, symptoms, laboratory results or diagnoses, and multiple observation records can occur at a single consultation. Consultation records do not contain such granularity, so observation records were selected for analysis. Drug issue records contain medications prescribed. Dementia-specific medications include prescriptions for four drugs advised for use by the NHS for PLWD [30]: Donepezil, Galantamine, Rivastigmine and Memantine. Non-dementia medications refer to all remaining drugs prescribed to the sample population. Rates per year of three patient-GP contacts types were calculated: dementia medications, non-dementia medications and observations.

Date of diagnosis is not specified in CPRD. Previous research using CPRD calculated date of diagnosis as the first GP observation record with the condition under investigation specified [31]. We defined date of diagnosis as the date of the individuals’ first dementia-specific observation, based on the any of the following diagnosis terms noted: “dementia”, “Alzheimer”, “cogniti”, or “memory”. These terms relate to specific codes entered into the GP system and reflect a patient presentation to GP. Using this date as date of diagnosis will both standardise diagnosis date across the sample population, and define the first date at which a dementia-specific event was observable within an individuals’ GP records. Diagnosis date was set as ‘year 0’ for each person in the study. We include only patient-GP contact records after the date of diagnosis.

Number of years people were present in the data were calculated by subtracting the year of their diagnosis, from the year of their final GP contact. Patient-GP contact rates were calculated by dividing their total GP contacts by the total years of GP-contact data available. Healthcare use variables (GP observations and dementia/non-dementia medication rates) have been included to measure an individual’s experiences with healthcare. Effective treatment and management of dementia and comorbidities has been illustrated to impact survival rates among PLWD [16,17]. High observations may also act as a proxy for health status, since individuals who require more observations may have greater needs (rather than only reflecting effective treatment). We use these three primary healthcare use variables as explanatory factors, to identify how use of such services and medications can impact mortality risk.

Individuals’ age at diagnosis- and dementia onset-sex and GP region were available from their GP records. Ethnicity was available from individuals’ secondary healthcare records. The 2011 GP urban/rural classification and 2019 Indices of Multiple Deprivation (IMD) quintile were available through data linkage. Inequalities in dementia and dementia outcomes have been illustrated across spatial contexts—including where somebody lives, the levels of deprivation in the area they live, and where their GP is based [17,32,33]. Such variations can exist due to the social gradient in wider determinants of health, differences in local funding and service equity, the make-up of local healthcare systems and regional differences in population composition [34,35,36,37]. As a result, GP region, urban/rural GP classification and IMD 2015 deprivation quintile were considered as explanatory factors for variance in mortality risk among PLWD.

### 2.4. Missing Data

Ethnicity was missing for 7421 (5.2%) people, and 276 (0.2%) people had no available IMD 2015 deprivation (Table 1). Statistical analyses including variables with missing data will remove those individuals with missing data.

### 2.5. Sample Population

Our analytical sample size was 142,340 people (Figure 1). Individuals with missing ethnicity or IMD quintile were not included regression models, with data assumed missing completely at random. We defined the sample population as patients registered at GPs in England, diagnosed with dementia between 2002–2016, with at least two years of GP follow-up data.

### 2.6. Statistical Analysis

Analysis was conducted in R, with descriptive statistics calculated to summarise the dataset. Inequalities in mortality risk were examined using Cox Proportional Hazards (CPH) regression models, with Bonferroni correction applied to derived *p*-values to account for the potential for Type I errors when calculating multiple, simultaneous statistical tests. CPH demonstrate simultaneous impact of multiple explanatory factors on the occurrence of an event (i.e., mortality). Separate models were applied to early- and later-onset dementia, generating hazard risk values for mortality risk by explanatory factors.

## 3. Results

### 3.1. Sample Population Characteristics

Two-thirds of the sample population are female (Table 2), less than 4% are of non-White ethnicities, ~80% are aged 75–94 years and a more live in the more deprived quintiles.

Older age groups and areas with increased levels of deprivation had the highest rates of mortality (Table 3). There were differences in mortality by geographic region; greatest mortality was in the North East (57.2%), and lowest in London (44.3%). Asian (43.6%), Black (40.0%) and Mixed/Other (41.9%) groups had lower mortality than White (51.5%).

### 3.2. Mortality Inequalities in Early-Onset Dementia

Regression analyses found few factors had a significant impact on mortality risk among those with early-onset dementia (Table 4). Accounting for covariates (age at diagnosis, ethnicity, IMD 2015 deprivation quintile, urban-rural GP classification, GP region and patient-GP contact rates), men had significantly greater mortality risk than women and, higher rates of GP observations were significantly associated with greater mortality risk.

### 3.3. Mortality Inequalities in Late-Onset Dementia

Regression analysis for later-onset dementia found distinct, significant demographic inequalities. When accounting for covariates (Table 4), men had significantly greater mortality risk than women. Age was significantly, positively associated with mortality risk, with each year of age associated with 4% (hazard risk (HR): 1.04; confidence Intervals: 1.04–1.05) greater likelihood of dying in the study. Significant variance in mortality risk was found among ethnic groups, with Black (HR: 0.71; 0.64–0.79) and Mixed/Other (HR: 0.75; 0.65–0.86) people having lower mortality risk than White people.

Deprivation was also significantly associated with mortality risk. Compared to the least deprived quintile (Quintile 1), each quintile had significantly increased mortality risk with a dose-response relationship; the most deprived quintile (Quintile 5) had a 20% increased risk of mortality (HR: 1.20; 1.15–1.24) compared to the least deprived. Significant differences were found across geographic regions. Compared to the South East Coast region, mortality risk was greater in the South Central (HR: 1.23; 1.17–1.29), South West (HR: 1.17; 1.11–1.23), North East (HR: 1.10; 1.03–1.16).

Rates of GP observations and dementia medications were also associated with significantly greater risk of mortality among people living with late-onset dementia.

## 4. Discussion

### 4.1. Key Findings

This study presents one of the first to apply large EHRs in exploring numerous causes of inequalities in dementia-related mortality. In early- and late-onset dementia, increasing rates of GP observation contacts—and for late-onset, dementia medications—are associated with increased mortality risk. Additionally, men with early-onset have higher mortality risk. For people with later-onset dementia, however, those of Black and Mixed/Other ethnicity had lower mortality risk. Additionally, the most deprived areas had greatest mortality risk and higher mortality was observed in North East, South Central and South West GP regions.

### 4.2. Research Context

Existing research examines the impact of few social, demographic or geographic factors in mortality risk among PLWD [1,2,40]. Yet none focus on multiple risk factors or investigate healthcare contacts as explanatory factors in health outcomes for PLWD. Despite continuing evidence of dementia care inequalities [17] and UK Strategies prioritising narrowing inequalities [4], issues persist, and have been exacerbated throughout the COVID-19 pandemic [18]. Our findings progress previous research, highlighting social, demographic, geographic and healthcare inequalities in mortality risk, accounting for a greater range of explanatory factors.

In areas of greatest deprivation, there are more diagnosed and undiagnosed PLWD [41]. People from the most deprived areas receive less, and inadequate care and treatment [22,42]. In dementia, this leads to later diagnosis, with PLWD from deprived areas more likely to experience faster disease progression, use emergency healthcare with more severe symptoms [43], have shorter survival and greater mortality risk [16].

People registered with North East, South Central and South East GPs had greater mortality risk. The North East—along with higher prevalence of poor health [44] and lowest life expectancy in England [45]—contains some of the most socio-economically deprived areas in England [46], likely contributing to higher mortality risk for. Greater dementia mortality risk in South Central and South East regions is reflective of demographic and systemic factors. The South of England tends to be more affluent, with greater life expectancy [47]. However, pockets of deprivation may be hidden in regional analyses. Greater mortality risk for older PLWD was observed, and White people had higher mortality risk than other ethnicities. With increased mortality risk observed in GP regions with varied levels of deprivation, mortality risk may be more likely a result of the regions’ population composition, rather than the geographic region itself. The South Central and South East regions are older and less ethnically diverse [48]. An older population, with reduced mobility and further to travel to access care, will have less frequent healthcare contact, limited treatment options, faster disease progression, and greater risk of negative health outcomes [49,50].

Black and Mixed/Other groups had lower mortality risk than White people; findings similar to UK dementia-related mortality [51]. White populations in the UK have lower life expectancy than other ethnic groups [52], reflective of minority ethnic groups having younger population demography [53]. Differences in mortality risk by ethnicity are associated with demographic, systemic and cultural factors. Cultural differences mean greater familial care provision for PLWD among minority ethnic communities. Familial care in ethnic minority communities can be seen as obligatory, and provided by multiple, younger family members, more capable of providing physical care, better support and enabling longer time at home [54].

### 4.3. Implications for Policy, Practice and Research

This study illustrates increasing frequency of patient–GP contact and rates of dementia medications are associated with greater mortality risk among PLWD. Previous research highlights later diagnosis, faster disease progression and greater mortality risk associated with differential healthcare use [23,55]. Different medications for dementia subtypes [56], inappropriate medications and primary healthcare use, and increased secondary healthcare contact, all impact outcomes [57,58,59]. PLWD also have high comorbidities, particularly older PLWD, emphasising how vital treating dementia and comorbidities is to maintaining health and quality of life [60,61]. Furthermore, holistic care, involving non-medicative treatments can be beneficial [62,63].

There is a growing need to improve understanding of health inequalities for PLWD [64]. This study illustrates the potential and importance of using existing EHRs to explore health inequalities. Further use of EHRs to evidence inequalities in various outcomes for PLWD can benefit policy-makers, service commissioners and providers and clinicians. Developing further on this study, future research should develop understanding of variation in primary and secondary healthcare use among PLWD. Exploring numerous dementia-specific, and non-dementia healthcare contact types is important to identify inequalities in the need and use of healthcare services among PLWD. Future research can extend this work through examining the healthcare pathways, and temporal service use changes among PLWD. For example, by clustering PLWD based on type and frequency of use of different healthcare services, one can identify social, demographic and geographic groups most likely to need and use different services. This can help us to explain why inequalities exist, building on more descriptive work exploring the extent of inequalities. Additionally, EHRs can be used to identify care pathways, helping to illuminate patterns in care usage and associations with positive or negative outcomes. Knowledge of the pathways more likely to elicit positive outcomes can guide future service provision and care decision-making.

### 4.4. Limitations

There are potential explanatory factors of outcome inequalities for PLWD which are not available through CPRD. There is a need to improve data collection for PLWD; routine data of EHRs should be more inclusionary and representative, aiming to be more complete in existing metrics and expanding social, demographic and geographic metrics recorded. There are issues around the representativeness of the sample population within CPRD. Although the population in our study is relatively representative of the UK dementia population for most social and demographic variables—as seen in Table 2 and Table 3—there are differences between the GP region of the sample population compared to the UK dementia population. Over/under-representation of certain regions may introduce selection bias into our analyses, which may limit the generalisability of our findings (especially the inequalities by region we identify). CPRD holds only data from GPs who have registered to send their practice data to CPRD. If those GPs engaging with data sharing are not random and socially/geographically patterned, this may contribute to the bias in our data.

Given the nature of the condition, date of dementia diagnosis is difficult to define. Though screening and testing can indicate dementia symptomology, there is a reliance on clinical judgement during healthcare interactions [65]. Lack of GP confidence in diagnosing, or lack of knowledge of dementia in primary care may result in issues around dementia diagnosis, resulting in issues with defining dementia date of diagnosis [66]. There is currently no way to negate this, however in standardising date of diagnosis in the sample population, adds a greater degree of precision.

Evidence points to greater prevalence of diagnosed and undiagnosed dementia among some minority ethnic groups [67], and in areas of greatest deprivation [68]. Furthermore, less reliance on formal care for PLWD among some communities [69], inadequate collection of socio-economic and demographic information and historic under-representation in EHRs and research among some communities [70], means true prevalence among certain population groups may not be discovered [71]. Potential lack of representation in CPRD may impact potential findings and conclusions.

For the sample population, there are fewer dementia medication records (~2.5% of all recorded patient-GP) than GP observation or non-dementia medication records. In early- and later-onset dementia there are fewer than 2.4 and 2.9 dementia medication records per patient year respectively. This may help explain wider confidence intervals for dementia medication contacts as an explanatory factor for mortality risk in the sample population.

## 5. Conclusions

Findings from this study suggest substantial differential mortality risk among PLWD, due to demographic, social and geographic factors, and use of primary healthcare. These findings have ramifications for future research and services. Reducing inequalities in mortality for PLWD requires systemic, societal and cultural measures. In areas of greatest deprivation, expansion of health and social care provision, alongside improved links between primary healthcare and post-diagnostic support, can make services more accessible. Additionally, commitment to person-centred care discussions are essential, with pragmatic inclusion of medicative and non-medicative treatments. Better access to, and support in using health technologies, alongside improved transport infrastructure can enable more equitable service access for remote and older populations. Future research should explore more health and social outcomes for PLWD. This study was the first to incorporate numerous socio-economic and geographic factors, and healthcare contacts as factors in mortality risk among PLWD. Future research should include broader healthcare contact types, and socio-economic and geographic characteristics as explanatory factors of health outcomes for PLWD.

## Figures and Tables

**Figure 1 ijerph-18-13405-f001:**
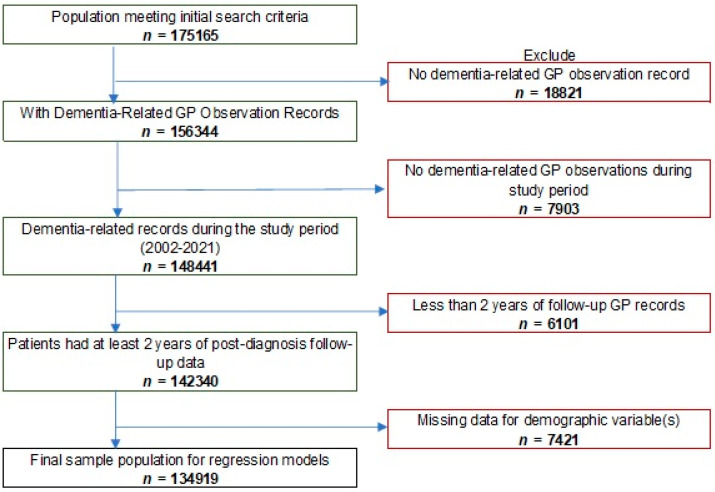
Sample population flowchart inclusion/exclusion criteria.

**Table 1 ijerph-18-13405-t001:** Available/missing explanatory variables data for sample population.

Group	Population (*n*)	% Present	Missing (*n*)
Total sample population	142,340	-	-
With ethnicity stated	134,919	94.8%	7421
With fields present to calculate age	142,340	100%	-
With gender stated	142,340	100%	-
With IMD 2015 deprivation quintile stated	142,064	99.8%	
With general practice (GP) region stated	142,340	100%	-
With urban–rural GP classification stated	142,340	100%	-

**Table 2 ijerph-18-13405-t002:** Demographics of UK dementia population vs. sample population.

Demographic	Study Cohort		UK ^1^
	*n*	%	%
Total study population	142,340		
Sex			
Female	94,060	66.1%	65.0%
Male	48,280	33.9%	35.0%
Age Group			
Under 45	104	0.1%	0.2%
45–54	870	0.6%	0.5%
55–64	4237	3.0%	4.5%
65–74	20,516	14.4%	16.6%
75–84	63,236	44.4%	36.5%
85–94	49,086	34.5%	36.2%
95+	4291	3.0%	5.5%
Onset			
Early (<65)	5211	3.7%	5.2%
Later (65+)	137,129	96.3%	94.8%
Urban/Rural GP Classification			
Urban	121,612	85.4%	
Rural	20,728	14.6%	
Ethnicity			
White	129,653	96.1%	94.5%
Asian	1939	1.4%	1.5%
Black	2247	1.7%	2.7%
Mixed/Other	1080	0.8%	1.3%
Missing	7421	5.2%	
IMD 2015 Deprivation Quintile			
1 (least deprived)	33,370	23.5%	26.5%
2	32,786	23.1%	23.4%
3	28,610	20.1%	20.5%
4	24,932	17.5%	16.6%
5 (most deprived)	22,366	15.7%	10.7%
Missing	276	0.2%	2.4%
GP Practice Region			
South East Coast	12,057	8.5%	17.3%
North East	7428	5.2%	5.3%
North West	25,427	17.9%	13.5%
Yorkshire And The Humber	6139	4.3%	9.9%
East Midlands	3020	2.1%	9.3%
East of England	8261	5.8%	11.8%
West Midlands	24,779	17.4%	10.2%
London	14,830	10.4%	11.0%
South Central ^2^	19,584	13.8%	-
South West	20,815	14.6%	11.7%

(^1^ UK prevalence by explanatory factors Alzheimer’s Research UK [38]. ^2^ GP regions for UK data based on GP regions from dementia prevalence estimates [39]).

**Table 3 ijerph-18-13405-t003:** Sample population mortality and available years of data (from year of diagnosis to date of final recorded GP contact/death), by socio-economic and geographic variables.

Group	Died	% Died	Total Data Years	Data Years Per Patient	Total Patients
Sex					
Female	47,655	50.7%	1,037,575	11.03	94,060
Male	25,114	52.0%	545,912	11.31	48,280
Dementia Onset					
Early Onset	1727	33.1%	59,550	11.43	5211
Later Onset	71,042	51.8%	1,523,937	11.11	137,129
Urban/Rural GP Classification					
Urban	62,120	51.1%	1,355,230	11.14	121,612
Rural	10,649	51.4%	228,257	11.01	20,728
Age Group					
Under45	29	27.9%	1133	10.89	104
45–54	278	32.0%	9564	10.99	870
55–64	1420	33.5%	48,853	11.53	4237
65–74	8001	39.0%	242,220	11.81	20,516
75–84	30,652	48.5%	720,722	11.40	63,236
85–94	29,234	59.6%	518,774	10.57	49,086
95+	3155	73.5%	42,221	9.84	4291
Ethnicity Group					
White	66,817	51.5%	1,443,890	11.14	129,653
Asian	845	43.6%	23,629	12.19	1939
Black	899	40.0%	26,023	11.58	2247
Mixed/Other	453	41.9%	12,114	11.22	1080
IMD 2015 Deprivation Quintile					
1: Least Deprived	16,404	49.2%	377,391	11.31	33,370
2	16,851	51.4%	364,830	11.13	32,786
3	14,841	51.9%	319,408	11.16	28,610
4	12,687	50.9%	274,378	11.01	24,932
5: Most Deprived	11,853	53.0%	244,494	10.93	22,366
GP Region					
South East Coast	5816	48.2%	136,116	11.29	12,057
North East	4252	57.2%	85,681	11.53	7428
North West	13,418	52.8%	289,817	11.40	25,427
Yorkshire And The Humber	3082	50.2%	68,595	11.17	6139
East Midlands	1410	46.7%	32,825	10.87	3020
East of England	4135	50.1%	91,006	11.02	8261
West Midlands	12,307	49.7%	275,161	11.10	24,779
London	6573	44.3%	165,344	11.15	14,830
South Central	10,695	54.6%	215,802	11.02	19,584
South West	11081	53.2%	223,140	10.72	20,815

**Table 4 ijerph-18-13405-t004:** Fully-adjusted for covariates. ^1^ Cox proportional hazards model for sample population with early- and later-onset dementia, by explanatory factors.

Group	Early-Onset Dementia				Later-Onset Dementia			
Sex	HR	95% CI	Adjusted *p*-Value ^2^		HR	CI (95%)	Adjusted *p*-Values	
Female	1.00				1.00			
Male	1.24 **	(1.09–1.41)	0.032	*	1.11 ***	(1.09–1.14)	0.000	***
Age								
Age At Diagnosis	1.00	(0.99–1.01)	1.000		1.04 ***	(1.04–1.05)	0.000	***
Ethnicity								
White	1.00				1.00			
Asian	0.64	(0.39–1.02)	1.000		0.80 ***	(0.72–0.89)	0.095	
Black	0.88	(0.54–1.43)	1.000		0.71 ***	(0.64–0.79)	0.000	***
Mixed/Other	1.18	(0.53–2.66)	1.000		0.74 ***	(0.65–0.86)	0.004	**
IMD 2015 deprivation quintile								
Quintile 1: Least Deprived	1.00				1.00			
Quintile 2	1.18	(0.97–1.45)	1.000		1.08 ***	(1.04–1.11)	0.012	*
Quintile 3	1.09	(0.88–1.35)	1.000		1.07 ***	(1.03–1.11)	0.000	***
Quintile 4	1.10	(0.89–1.38)	1.000		1.09 ***	(1.05–1.13)	0.055	
Quintile 5: Most Deprived	1.06	(0.85–1.34)	1.000		1.20 ***	(1.15–1.24)	0.000	***
Urban/Rural GP Classification								
Urban	1.00				1.00			
Rural	0.96	(0.78–1.16)	1.000		1.01	(0.97–1.04)	1.000	
GP Region								
South East Coast	1.00				1.00			
North East	0.96	(0.65–1.43)	1.000		1.10 **	(1.03–1.16)	0.000	***
North West	1.09	(0.81–1.48)	1.000		1.04	(0.99–1.09)	0.000	***
Yorkshire And The Humber	1.22	(0.80–1.87)	1.000		1.06	(0.99–1.13)	0.962	
East Midlands	0.97	(0.60–1.57)	1.000		1.03	(0.94–1.13)	1.000	
East of England	0.96	(0.63–1.47)	1.000		1.06	(0.99–1.13)	1.000	
West Midlands	0.88	(0.64–1.22)	1.000		1.03	(0.98–1.08)	1.000	
London	1.09	(0.77–1.52)	1.000		0.95	(0.90–1.00)	1.000	
South Central	1.39 *	(1.02–1.88)	0.727		1.23 ***	(1.17–1.29)	0.000	***
South West	1.30	(0.95–1.78)	1.000		1.17 ***	(1.11–1.23)	0.000	***
Patient-GP Contact rates per year/100								
Observations	1.67 ***	(1.44–1.92)	0.000	***	1.94 ***	(1.91–1.97)	0.000	***
Dementia Medications	6.44 **	(1.73–23.69)	0.112		21.48 ***	(17.62–26.19)	0.000	***
Non-Dementia Medications	0.86 *	(0.78–0.98)	0.613		0.84 ***	(0.82–0.85)	1.000	

^1^ Covariates accounted for: age, sex, ethnicity, deprivation quintile, urban/rural GP, GP region and healthcare contacts; ^2^ Adjusted *p*-values for from Cox proportional hazards with Bonferroni adjustments applied; significance level codes: *** 0.001 (99.9%); ** 0.01 (99%); * 0.05 (95%).

## Data Availability

Data is not publicly available.

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
