# Peer review of "The Impact of Demographic, Socio-Economic and Geographic Factors on Mortality Risk among People Living with Dementia in England (2002–2016)"

_ijerph, 2021, doi:10.3390/ijerph182413405_

Round 1

Reviewer 1 Report

Does the introduction provide sufficient background and include all relevant references?

Yes although perhaps the change in diagnostic rates across time doesn’t receive attention. There could be more on proximity to death, with some pretty obvious findings such as those relating to rate of GP observation being associated with mortality – they’re getting sicker.

Is the research design appropriate? Yes

Are the methods adequately described? Perhaps there could be more description of the proportions of data from the different regions, as it seems there will be relatively small numbers and a huge number of comparisons (but good that Bonferroni correction applied).

Are the results clearly presented? Can be improved – it is worth going through the text and the tables carefully to make sure it’s all clear as some of the paper is a little shorthand and some of the tables have labels that are difficult to understand (on the principle that all tables should be interpretable even without having to refer to the text). Comparison with ‘UK dementia’ in first table merits further investigation – do you know how those UK data were estimated? It can sometimes be a bit circular. Given the findings it would be good if a coastal analysis could be done.

Are the conclusions supported by the results? Yes, perhaps the authors should consider a structured discussion – brief resume of key findings, critique of results with size of bias when thought to be present (eg differential diagnostic rates  in areas of disadvantage), then discuss the results that make it through the critique in the context of the literature provided in the background, then implications for research. practice, policy followed by conclusions.

English language and style are mostly fine although some further clarification to make the paper clearer will help the readers.

This is an interesting paper focused on people who have a set of dementia relevant observations in the notes, including some that are not dementia itself. The findings reinforce earlier findings related to substantial variation in experience of mortality. While there are some improvements to the paper as noted above it is well written and carefully presented.

Author Response

Thank you for taking time to read and review our research paper. We appreciate the constructive feedback and take on-board the points you have made in our revised paper. Please find our responses to your comments in the attached Word document.

Reviewer 2 Report

Thank you for an excellent report. The quantitative study outcomes provide valuable insight into the inequalities faced by PLWD with huge potential for ongoing research and health service improvement. Your introduction provided a very good background with appropriate references. The methods (data collection and analysis) are well suited to your aim and have been described very clearly. The results are well presented and align nicely with your aim - the statistical analyses are well executed, relevant and meet the needs of your study. The discussion provides good detail around your own study, shows high level thinking in relation to the current healthcare context and previous research. It would be beneficial to add a sentence about how a specific or planned future research within this population. There are some minor inconsistencies in your reference list (e.g. Reference number 30).  

Author Response

(The authors gave the same response as above.)

Reviewer 3 Report

The article demonstrates the relationship between mortality risk among people living with dementia(PLWD)and demographics, socioeconomic status, and geography. It also provide a basis for decision-making to eliminate inequalities in dementia diagnosis, support, treatment, and health care.

This research based on big data collected from electronic health records from a cohort of Clinical Practice Research Datalink (CPRD) General Practices (GPs) patients in UK. The statistics analysis used an appropriate model: Cox Proportional Hazards (CPH) regression. Therefore, the results are generalisable to a certain extent, which provides new evidences of inequalities in mortality risk of late-onset dementia socioeconomic, demographic and geographic factors, and frequency of GP contact.

However, there are still some problems about details as follows:

Firstly, the rationale of the Introduction part. “Introduction” mentioned comorbidities and the increasing number of PLWD, and these two aspects might cause inequalities in dementia support. However, what is still confusing is that these two parts still seem to emphasize the severity of dementia and struggling services, but there is little mention of why this research focus on  demographics, socioeconomic, and geographic factors. It needs more logic and arguments to support that why this topic was chosen.

In addition, the explanation of one of the three factors----“geographic factor” ----is not clearly stated. The division of "geographical area" is based on the 10 areas of England in the GPs region, but what are the specific differences in "geographical aspects" of these 10 areas? Is it only related to the distance and accessibility of medical services? This part should be explained. Moreover, the "Discussion" part of this article also mentioned that the results show that different regions have different death risks, which may be due to socioeconomic and ethnic rather than geographic factors. So whether "geographical factors" themselves can affect the death rate of dementia, the conclusion is not sure.

Also, there is a logical problem in the discussion part. There seems to be a wrong causal relationship between "low life expectancy for white British people" and "a younger minority/black population".

Author Response

(The authors gave the same response as above.)
